# Prevalence and Clinical Significance of Heart Murmurs Detected on Cardiac Auscultation in 856 Cats

**DOI:** 10.3390/vetsci9100564

**Published:** 2022-10-13

**Authors:** Luca Ferasin, Heidi Ferasin, Altin Cala, Naomi Creelman

**Affiliations:** 1Specialist Veterinary Cardiology Consultancy Ltd., Four Marks GU34 5AA, UK; 2The Ralph Veterinary Referral Centre, Marlow SL7 1YG, UK

**Keywords:** heart murmur, cats, feline, cardiology, SAM, DRVOTO, congenital, acquired, innocent, iatrogenic murmur

## Abstract

**Simple Summary:**

Heart murmurs are a common finding in cats and, in many cases, the presence of an audible murmur on cardiac auscultation does not necessarily imply the presence of an underlying heart disease. Several studies have been published in the past to address the prevalence and clinical significance of heart murmurs in cats, but very few have looked into the exact origin of the blood flow turbulence responsible for this finding. We retrospectively reviewed clinical records and echocardiographic examinations of 856 cats with heart murmur and found that the majority of murmurs are caused by systolic anterior motion of the mitral valve (SAM) (39.2%) and dynamic right ventricular outflow tract obstruction (DRVOTO) (32%). These causes of murmur do not appear associated with a structural cardiac abnormality in 56.1% (SAM) and 85.0% (DRVOTO) of murmurs. This study also demonstrated that some heart murmur characteristics (timing, intensity and location) can occasionally discriminate between normal cats and cats with underlying heart disease, with the exception of loud and palpable murmurs, which are inevitably associated with significant cardiac abnormalities. However, since the majority of heart murmurs in cats appear to be systolic and mild–moderate in loudness, echocardiography should always be considered following identification of a heart murmur on physical examination in cats.

**Abstract:**

Background: Cardiac auscultation is one of the most important clinical tools to identify patients with a potential heart disease. Although several publications have reported the prevalence of murmurs in cats, little information is available in relation to the exact origin of the blood flow turbulences responsible for these murmurs. The aim of this study was to determine the prevalence and clinical significance of murmurs detected during physical examination in cats. Methods: Retrospective evaluation of clinical records and echocardiographic examinations performed in cats for investigation of heart murmurs; Results: Records of 856 cats with full clinical information were available for review. The cause of murmur was identified in 93.1% of cases (72.3% with single blood flow turbulence, 26.4% with two, and 1.3% with three identifiable sources of murmur). Systolic anterior motion of the mitral valve (SAM) was the primary cause of murmur in this population (39.2%), followed by dynamic right ventricular outflow tract obstruction (DRVOTO) (32%) and flow murmurs (6.9%). Most cats with a murmur (56.7%) did not present any structural cardiac abnormality. Conclusions: This study indicates that some heart murmur characteristics (timing, loudness and point of maximal intensity) can potentially predict the presence of an underlying cardiac disease.

## 1. Introduction

In people, heart murmurs are the most common reason for referral to a cardiologist and, especially in children, approximately 50–70% of these murmurs are clinically insignificant [1]. Similarly, the discovery of a heart murmur is a common justification for referral in feline medicine and, based on our clinical case log, approximately 50% of all cats seen at our institutions are referred for further investigation of a heart murmur. Another analogy with human cardiology is that many heart murmurs are detected in apparently healthy cats and cardiac disease is absent in as many as 50% of cats with heart murmurs [2,3]. However, many published studies on heart murmurs in cats have been primarily focusing on reporting the underlying structural abnormalities (e.g., cardiomyopathy, left ventricular hypertrophy, among others) rather than identifying the exact source of the blood flow turbulence [2,3,4,5,6,7], although an attempt to report the correct etiology of the murmur was reported in two of these studies [2,6].

Therefore, the main aim of this retrospective study was to determine the prevalence of heart murmurs detected during routine clinical examination in cats, with a particular emphasis on obtaining the exact echocardiographic identification of the origin of the flow turbulence deemed responsible for those murmurs. This study also sought to potentially identify patient and murmur characteristics that might predict a diagnosis of significant congenital or acquired cardiac disease.

We hypothesized that the majority of cats with a heart murmur detected on cardiac auscultation do not have an identifiable cardiac abnormality on echocardiographic examination as previously observed in other studies [2,3] and that some patient (age, breed, gender and body weight) and heart murmur characteristics (timing, intensity, PMI, dynamic or intermittent nature) may predict the presence of a structural cardiac abnormality.

## 2. Materials and Methods

### 2.1. Data Acquisition

Clinical records and echocardiographic examinations performed on cats between June 2009 and June 2022 for diagnostic investigation of a heart murmur at 2 institutions, namely Specialist Veterinary Cardiology Consultancy (SVCC Ltd., Four Marks, Hampshire, UK) and The Ralph Veterinary Referral Centre (TRVRC Marlow, Buckinghamshire, UK), were reviewed retrospectively. The relevant information included examination date, cat’s signalment, heart murmur characteristics, echocardiographic identification of blood flow turbulence (single or multiple) associated with the detected heart murmur, echocardiographic measurements, presence of structural abnormalities, electrocardiographic (ECG) findings and whether or not the cat underwent sedation prior to the echocardiographic examination. Cats were excluded from the study if they had incomplete clinical information, including the full description of the detected heart murmur or if the full native echocardiographic examination was not available for review.

### 2.2. Heart Murmur Characteristics

Essential description of heart murmurs included timing (systolic, diastolic, continuous and to-and-fro), loudness/intensity (Levine grading scale 1/6) [8], point of maximal intensity (PMI), classified as left and right parasternal, sternal/both parasternal, left base, and left apex. Murmurs varying in intensity during auscultation were described as “dynamic”, while murmurs that were not consistently audible were defined as “intermittent”. Cardiac auscultation was always performed in a quiet consult room by the attending cardiologist (LF or HF) in order to confirm the presence of a heart murmur and describe the murmur characteristics prior to the echocardiographic examination and always performed using the same model of stethoscope (Littman Classic II Pediatric stethoscope, 3-M, Maplewood, MN, USA).

### 2.3. Echocardiographic Examination

All echocardiographic examinations were performed by a board-certified cardiologist (LF) or a RCVS certificate holder in veterinary cardiology (HF) using an Esaote MyLab 30 Gold Cardiovascular or MyLab Omega (MyLab, Esaote S.p.A. Genoa, Italy) (SVCC Ltd.), or Esaote MyLab X8 (MyLab, Esaote S.p.A.) (TRVRC Genoa, Italy) echocardiographic machine and a 7.0–10.0 MHz or a 2.0–9.0 MHz phased array transducer depending on the size of the cat. All echocardiographic examinations were reviewed using dedicated offline ultrasound imaging software (MyLab Desk, Genoa, Italy and MyLab Desk Evo Genoa, Italy, Esaote S.p.A.) to verify the correct identification of significant blood flow turbulence and confirm the final diagnosis.

Complete transthoracic echocardiographic examination (2-dimensional, M-mode, color and spectral Doppler studies) was performed using routine echocardiographic views [9]. All cats were gently restrained in both right and left lateral recumbences on a padded echocardiographic table with their thoracic limbs moved slightly cranially to allow access to the relevant intercostal spaces. A small area between the fourth and fifth intercostal spaces was clipped before the application of ultrasonographic gel to improve image quality. Light sedation was administered when deemed necessary in some uncooperative cats.

Simultaneous ECG monitoring was obtained by applying non-traumatic ECG clips just above or below the elbows and just above or below the patella, using ultrasound gel as a conductive medium [10]. Whenever possible, echocardiographic measurements were taken during periods of sinus rhythm.

Left ventricular dimensions were measured from the right parasternal short axis view at the level of the papillary muscles using B-mode images from previously acquired high frame-rate cine-loops. For the purposes of this study, left ventricular hypertrophy was defined as a measurement of either interventricular septum in diastole (from leading to trailing edge) or left ventricular free wall in diastole (from leading to leading edge) greater than 6.0 mm, avoiding prominent false tendons, focal myocardial thickening at their insertion or punctual hyperechogenicity [11,12]. Left atrial size (LA) was assessed by indexing LA diameter to the diameter of the aorta using a right parasternal short axis view at the level of the heart base in early diastole at the first frame after aortic valve closure and left atrial enlargement was defined as LA diameter to the diameter of the aorta ≥ 1.5. All measurements were taken over three consecutive cardiac cycles. The origin of the heart murmur was obtained by observing a significant blood flow turbulence, initially identified by color flow mapping and, subsequently, on Doppler spectral flow analysis.

### 2.4. Diagnostic Criteria

#### 2.4.1. Systolic Anterior Motion of the Mitral Valve (SAM)

Diagnosis of SAM was characterized by the concurrent presence of motion of the septal mitral valve leaflet towards the interventricular septum during systole, observed on a right parasternal short-axis M-mode view of the left ventricle at the level of the mitral valve, turbulent flow in the left ventricular outflow tract with simultaneous mitral regurgitation in mid-systole systole (typical ‘double jet’) and a dagger-shaped (scimitar-like) flow profile on spectral Doppler interrogation of the left ventricular outflow tract [13,14,15,16]. Cases of systolic anterior motion of chordae tendineae were included in this group [17]. Inducible SAM was defined as a dynamic outflow obstruction induced by an increased sympathetic stimulation evoked by a sudden increased loudness of the Doppler sound through the speakers of the ultrasound machine (provocative maneuverer).

#### 2.4.2. Dynamic Right Ventricular Outflow Tract Obstruction (DRVOTO)

The presence of DRVOTO was confirmed as systolic color Doppler aliasing in the right ventricle (RV) just cranial to the tricuspid valve and extending into the RV outflow tract, associated with spectral dispersion and a scimitar-like appearance of the RV outflow profile on spectral Doppler study, as previously described [18,19]. Inducible DRVOTO was described as a blood flow turbulence obtained by applying a gentle pressure with the ultrasound transducer to the right parasternal area of the cat, inducing a iatrogenic dynamic outflow obstruction [18].

#### 2.4.3. Flow Murmurs

Diagnosis of flow (often named “innocent”) heart murmurs was based on the absence of any detectable structural or functional abnormalities on echocardiographic examination, despite the presence of a soft heart murmur on auscultation [20,21].

#### 2.4.4. Mitral and tricuspid valve regurgitation and stenosis

Mitral valve and tricuspid valve regurgitation were diagnosed following identification of a systolic turbulence directed towards the left and right atrium, respectively. The valvular regurgitation was confirmed in the left apical view using spectral Doppler interrogation of the abnormal flow. Conversely, mitral valve stenosis was diagnosed by demonstrating a diastolic turbulence directed towards the left ventricle and a typical prolonged deceleration time of the E wave on spectral Doppler inflow studies [22].

#### 2.4.5. Mid-Left Ventricular Outflow Obstruction

Mid-left ventricular outflow obstruction, also called mid-cavitary outflow obstruction, was defined as a systolic turbulent flow originating at the middle of the left ventricle as a result of segmental mid-septal hypertrophy and misplacement of the papillary muscles and chordae tendineae in cats with left ventricular hypertrophy [23,24].

#### 2.4.6. Pulmonic and Aortic Stenosis and Insufficiency

Pulmonic stenosis (PS) and aortic stenosis (AS) were deemed to be valvular unless otherwise stated and were confirmed by identification of partially fused cusps and increased systolic blood flow across the pulmonic and aortic valve respectively. Sub-aortic stenosis and supravalvular aortic stenosis were diagnosed after identification of a systolic blood flow turbulence in correspondence of a membrane just below or above the aortic valve respectively, causing a fixed obstruction. Double-chambered right ventricle (DCRV) was diagnosed when the blood flow turbulence and associated increased flow velocity was originating from a mid-cavitary obstruction dividing the right ventricle into a high-pressure proximal portion and a low-pressure distal portion. Aortic insufficiency (AI) was diagnosed after identification of a significant diastolic flow from the aorta towards the left ventricular cavity, mostly associated with aortic valve endocarditis or aortic valve dysplasia [25,26].

#### 2.4.7. Ventricular and Atrial Septal Defects (VSD and ASD)

Heart murmurs associated with VSD and ASD were diagnosed following visualization of an incomplete septum and identification of a systolic flow turbulence across the defect [27].

#### 2.4.8. Tetralogy of Fallot

Blood flow turbulence associated with tetralogy of Fallot was characterized by a combined pulmonic stenosis and right-to-left shunting VSD [27].

#### 2.4.9. Patent Ductus Arteriosus (PDA)

Diagnosis of patent ductus arteriosus (PDA) was based on the identification of the ductus arteriosus and an associated continuous blood flow turbulence across the defect directed toward the main pulmonary artery, since all PDAs in this cohort were left-to-right shunting [28].

#### 2.4.10. Atrioventricular Canal Defect

The blood flow turbulence associated with atrioventricular canal defect, also called cushion defect, was mainly characterized by a systolic shunt of the blood passing from the left ventricle to the right ventricle and out the pulmonary artery [29].

#### 2.4.11. Coronary Artery to Pulmonary ARTERY fistula

Heart murmur in coronary artery to pulmonary artery fistula was associated with a continuous left-to-right blood flow turbulence across the abnormal vascular communication [30].

### 2.5. Statistical Analysis

Collected data were transferred to an electronic spreadsheet (Microsoft Office Excel 365, Microsoft Corporation, Redmond, DC, USA) and verified by 3 investigators (LF, NC and AC) for accuracy. Quantitative variables were analyzed for normal distribution using the Shapiro–Wilk test and subsequently reported as mean ± standard deviation if normally distributed or median and range if non-normally distributed. Qualitative variables were summarized using absolute and relative frequencies.

To test the hypothesis that in the majority of cats with a heart murmur detected on cardiac auscultation the murmur was not associated with a structural cardiac disease, we simply calculated the percentage of cats with a heart murmur without echocardiographic evidence of structural heart abnormalities.

Univariable logistic models were used to test the hypothesis that patient signalment (age, gender, breed, body weight) and heart murmur characteristics (timing, intensity, PMI, dynamic or intermittent nature) can predict the presence of a structural cardiac dis-ease. For this purpose, cat’s age, gender, breed as well as heart murmur characteristics were assessed as categorical variables, and murmur intensity was reclassified as “soft” (grade 1/6 or 2/6), “moderate” (grade 3/6), “loud” (grade 4/6) or “palpable” (grade 5/6 and 6/6), following the scheme proposed by Rishniw in 2018. This system is described as a simpler and more intuitive grading scheme that should provide the same clinical information as that obtained with the more complex Levine scheme but with potentially less confusion [8]. Multi-variable logistic regression analysis was used to determine whether each variable that was significant in the univariable analysis was significantly associated with an underlying structural cardiac abnormality. A *p* value of <0.05 was considered to be statistically significant. All statistical analysis was performed with a commercially available statistical program (MedCalc Software Ltd., Ostend, Belgium).

### 2.6. Ethical Approval

Ethical approval was not sought due to the non-invasive and retrospective nature of this study.

## 3. Results

### 3.1. Population Characteristics

The database query identified a total of 1521 cats that were referred for heart murmur investigation between June 2009 and June 2022. However, only 856 of these cases had full clinical information available for review, including all native echocardiographic images and video clips. All patients were deemed to have been clinically stable to undergo a full echocardiographic examination.

#### 3.1.1. Animals

The median age of cats selected for this study was 5.6 years (range, 1 month–19.5 years). The median body weight was 4.4 Kg (range, 0.3–12.4 Kg). There were 539 males (63.0%) and 317 females (37.0%), of which 463 were castrated males (54.1%) and 267 were spayed females (31.2%). Domestic shorthair (DSH) cats were the most represented breed in this study (60.5%), followed by domestic longhair (DLH) cats (6.8%) and British shorthair cats (6.1%). The number and percentage of remaining breeds are reported in Table 1.

#### 3.1.2. Sedation

A total of 73 cats (8.5%) were sedated prior to their echocardiographic evaluation due to suboptimal patient compliance or because they were already sedated for other diagnostic procedures not necessarily related to their cardiac investigation. Different protocols were used over the years based on clinician’s personal preference, including combined injectable ketamine/midazolam, combined injectable butorphanol/acepromazine, combined injectable butorphanol/alfaxalone, injectable buprenorphine or oral gabapentin.

### 3.2. ECG Findings

The majority of cats displayed sustained normal sinus rhythm (91.4%). Occasionally, sinus rhythm was observed in combination with ventricular ectopics (3.8%) or atrial premature complexes (2.1%). Sustained arrhythmias and conduction abnormalities included atrial fibrillation (1.0%), isorhythmic atrioventricular dissociation (0.9%), ventricular tachycardia (0.4%), atrial standstill (0.2%) and complete atrioventricular block (0.2%).

### 3.3. Heart Murmur Characteristics

Timing: the majority of heart murmurs detected in this feline population were systolic (98.7%), with only a small percentage of diastolic (0.7%), continuous (0.5%) and to-and-fro murmurs (0.1%). Diastolic murmurs were observed in cases of aortic insufficiency (AI), cor triatriatum sinister and mitral valve stenosis. However, not all cats with a significant AI had an audible diastolic murmur and this was considered an incidental finding when another cause of murmur, mainly systolic, was identified on echocardiography. Continuous murmurs were primarily associated with left-to-right shunting patent ductus arteriosus (PDA) (4 cats) or coronary artery to pulmonary artery fistula (1 cat), while the only to-and-fro murmur was reported for a case of mitral valve dysplasia with concomitant valvular regurgitation and stenosis.

Intensity: the majority of cats displayed a soft murmur (56.7%), followed by moderate (28.7%), loud (12.6%) and palpable (2.1%) (Figure 1), with the loudest murmurs always associated with a variety of severe congenital conditions (VSD, PDA, DCRV, PS and MV Dysplasia), occurring either alone or in combination. Approximately a quarter of the heart murmurs were reported to be intermittent (27.8%), and a similar number of murmurs were classified as dynamic (27.3%), where the change in loudness appeared associated with a change in heart rate (spontaneous or stimulated) or was induced by chest compression with the stethoscope [31].

Point of maximal intensity (PMI): this was described as sternal (or “both parasternal”) in 38.3% of cases, followed by left parasternal (37.2%), right parasternal (21.7%), left base (2.0%) and left apex (0.8%).

### 3.4. Cause of Heart Murmurs

The echocardiographic examination identified the cause of heart murmur in 93.1% of cases, based on a significant blood flow turbulence identified during color and spectral Doppler evaluation. A single significant blood flow turbulence source associated with the audible heart murmur was identified in the majority of cases (72.3%), while 226 cats (26.4%) and 11 cats (1.3%) had 2 or 3 identifiable causes of murmur, respectively, for a total of 1093 significant blood flow turbulences identified as causes of heart murmur in the 856 cats of this study.

#### 3.4.1. Heart Murmurs in Kittens

Heart murmurs were detected in 30 young kittens (less than 4 months of age), mostly identified after their first vaccination, and were associated with a congenital disease in 12 cases (40.0%). The remaining cases of heart murmur in this subgroup were associated with flow murmurs (60.0%). The youngest kittens diagnosed with an acquired cardiac disease (Hypertrophic Cardiomyopathy [HCM]-phenotype) were 6 months old and, in both cases, the echocardiographic changes were associated with suspected acute myocarditis, which was followed by a complete reverse cardiac remodeling (confirmed on repeat echocardiographic examination) after a few weeks and complete disappearance of their heart murmur.

#### 3.4.2. Most Common Causes of Heart Murmur

Systolic anterior motion of the mitral valve (SAM) was the most common cause of heart murmur identified in this feline population (39.2%), followed by dynamic right ventricular outflow tract obstruction (DRVOTO) (32%) and flow murmurs (6.9%). All different causes of blood flow turbulence responsible for heart murmurs in this feline population are reported in Table 2 and Figure 2.

#### 3.4.3. Association between Heart Murmurs and Structural Cardiac Abnormalities

The majority of cats that underwent full echocardiographic examination to identify the cause of a previously detected heart murmur did not present any structural cardiac abnormality (*n* = 485; 56.7%). These were mainly cases where the blood flow turbulence was associated with SAM (50.3%), DRVOTO (19.9%) and flow murmurs (14.6%). However, when these types of murmurs were analyzed individually as a sole cause of murmur, 56.1% of cats with SAM and 85.0% of cats with DRVOTO did not display any structural cardiac abnormalities on echocardiographic examination. None of the cats with flow murmurs had structural abnormalities, by definition. Most cats with DRVOTO were older individuals, with a median age of 9.3 years (range, 6 months–19.5 years).

The remaining causes of heart murmur were attributed to either congenital defects (14.0%) or acquired cardiac conditions (29.3%), with a relative frequency depending on the underlying condition, as reported in Table 2. All cats from this study with VSD or ASD showed a left-to-right shunting flow across the defect, with the only exception of Tetralogy of Fallot, characterized by a right-to-left shunt across the VSD. Interestingly, only 19.7% of cats with SAM had echocardiographic evidence of left ventricular hypertrophy.

Physical characteristics of the cats (age, gender, breed and bodyweight) and heart murmur characteristics (timing, intensity, PMI, dynamic or intermittent nature) were analyzed using univariable and multi-variable logistic regression analyses to assess the probability of identifying a structural heart disease on echocardiography. All palpable, continuous, diastolic and to-and-fro murmurs were associated with a significant congenital cardiac disease (100%). Conversely, a negative association was found between soft/moderate systolic heart murmurs, intermittent heart murmurs, murmurs with a PMI over the right parasternal area and the presence of a cardiac abnormalities on echocardiography (Table 3).

#### 3.4.4. Inducible Heart Murmurs

A group of cats with an audible heart murmur on cardiac auscultation did not reveal any significant blood flow turbulence on echocardiographic examination until a provocative maneuver was performed and such murmurs were classified as inducible heart murmurs (93 cats, 10.9%). These were characterized by inducible DRVOTO (8.3%) and inducible SAM (2.6%). In more detail, a provocative maneuver was necessary to reveal DRVOTO in 20.3% of cats and SAM in 5.1% of cats, respectively.

## 4. Discussion

To the best of the authors’ knowledge, this is the largest and most comprehensive study looking into the etiology and clinical characteristics of heart murmurs in cats and, unlike most of the published research on feline heart murmurs, is the first attempt to systematically identify the precise origin of the blood flow turbulence responsible for these murmurs.

The median age and body weight of cats referred to our clinics for cardiac investigation were very similar to other studies performed in cats referred to a cardiologist for further cardiac investigations [2,6].

The majority of murmurs detected on auscultation were systolic, soft or moderate and with a PMI over both parasternal areas (or simply “sternal”) with no apparent association with the underlying heart murmur etiology or the severity of the underlying heart condition. This suggests that heart murmur characteristics in the majority of cats may have a limited diagnostic relevance, although loudest murmurs are more likely associated with a pathological condition, mainly severe congenital cardiac defects, as previously reported [32]. The significance of intermittent heart murmurs may have different explanations. For example, these murmurs may be unmasked by an increased sympathetic tone due to stress or excitement, which ultimately increases heart rate, cardiac contractility, and blood pressure. Another possible explanation is the presence of inducible (iatrogenic) murmurs caused by overzealous chest compression with the stethoscope [31]. Similar phenomena can be hypothesized to explain the genesis of a dynamic heart murmur, with the only difference that dynamic murmurs are always audible, although they change in intensity during cardiac auscultation.

Echocardiographic identification of the blood flow turbulence responsible for the detected heart murmur was obtained in 93.1% of cases. In the remaining 6.9% of cases, echocardiographic examination failed to identify any significant turbulence or structural abnormality, which led to an echocardiographic diagnosis of flow (innocent) murmurs, adopting the same diagnostic criteria described in humans [20,21].

Approximately one quarter of cats had multiple blood flow turbulences identified on echocardiography, which were deemed responsible for the murmur detected on auscultation. This is, in our opinion, another important finding that reinforces the above observation that detection of heart murmurs in cats, in the absence of an echocardiographic examination, cannot provide sufficient information towards an etiological diagnosis, with the exception of intermittent heart murmurs inducible by chest compression and unmasked by varying the pressure of the stethoscope on the chest wall [31] or heart murmurs detected in young kittens, where loud murmurs are almost inevitably associated with a significant congenital defect (40.0% of cases) and soft murmurs mostly consistent with a benign (innocent) condition (60.0% of cases).

Unlike other studies where heart murmurs in cats appeared to be mostly associated with a cardiac disease [2,3,5,7], these results seem to indicate that over half (56.7%) of heart murmurs in cats are not associated with any structural cardiac abnormality.

The most common origin of heart murmurs in these cats was dynamic left ventricular outflow obstruction associated with SAM, which represents approximately 40% of all murmurs detected in this feline population. Eighty percent of cats with HCM-phenotype and a heart murmur on auscultation had SAM. Systolic anterior motion of the mitral valve was also present in other forms of cardiomyopathy, such as in 27.3% of restrictive cardiomyopathy (RCM)-phenotype, 33.3% of non-specific cardiomyopathy and 22.2% of end-stage HCM, based on the classification proposed by Luis Fuentes et al. [33], as well as in 24.2% of cats with mitral valve dysplasia. However, in 56.1% of all SAM cases, there was no echocardiographic evidence of structural heart disease, demonstrating that this form of dynamic left ventricular tract outflow obstruction is not pathognomonic of HCM, supporting the notion initially speculated, and subsequently confirmed, by Ferasin et al. [13,14].

The second most common cause of heart murmur in this feline population was represented by a dynamic right ventricular outflow tract obstruction (DRVOTO), which was identified in approximately one third of cases. DRVOTO did not seem to be associated with any cardiac disease in 85.0% of cases, and since the echocardiographic abnormalities observed in the remaining 15.0% of cases did not appear to affect the right outflow tract, we believe that the presence of DRVOTO in such cases represents a pure incidental finding.

Flow (Innocent) heart murmur represented the third most common cause of murmurs in this study (6.9%). However, it may be possible that the number of innocent murmurs was overestimated since a provocative maneuver was not performed in all echocardiographic examination to reveal the presence of inducible SAM or iatrogenic DRVOTO [18]. It is also possible that the diagnosis of a heart murmur on auscultation was occasionally inaccurate, such as in case of rapid respiratory sounds coinciding with the heart beat and subsequently misinterpreted as a heart murmur.

Blood flow turbulences associated with various congenital defects were considered responsible for an audible heart murmur in the remaining 21.9% of cases. However, it was impossible from our data to establish whether ASDs were responsible for an audible heart murmur since they were always associated with other significant blood flow turbulences.

The importance of this finding is particularly relevant for the clinical management of these cases, where an early diagnosis may change the clinical outcome by prompting corrective or palliative interventions (i.e., PDA, PS, AS) or regular monitoring of the sleeping respiratory rate to allow early identification of clinical signs associated with congestive heart failure [34,35].

Owing to the retrospective nature of this study, some data were missing in a number of patients, including blood pressure measurement, hematocrit and serum concentrations of thyroxine hormone, which may have had a potential impact on the genesis of the heart murmurs. Given the lack of data on systemic status of cats, it is not possible for the authors to conclusively exclude that, at least in some patients, the actual main or concurrent trigger of the murmur could have been misdiagnosed.

Another limitation was the inability to determine if outflow obstructions (namely SAM and DRVOTO) could have been potentially induced in some cases of “innocent” murmurs since the provocative maneuvers explained above were not always performed, especially when these techniques were still unknown or poorly described. Furthermore, limited characterization of the type of murmur for each echocardiographic abnormality may also represent a possible bias.

Another potential limitation is represented by the unknown effect of the various sedative protocols on the structural and functional echocardiographic findings, although a relatively small percentage of cats were sedated in this study. Similarly, information about the use of cardioactive drugs was missing in many cases, and therefore it was not possible to analyze their effect on the clinical and echocardiographic findings. Finally, different clinicians were involved in the diagnosis and management of cats included in this study and this could have affected the accuracy of the results, although all clinical records and echocardiographic examinations were retrospectively reviewed by a board certified cardiologist to double-check the accuracy of measurements and diagnoses.

## 5. Conclusions

More than half of cats referred for a cardiac evaluation of an audible heart murmur on thoracic auscultation do not have any echocardiographic evidence of heart disease, suggesting that most heart murmurs in cats are benign or may be potentially associated with a subclinical form of heart disease not detectable echocardiographically. This study also demonstrates that some heart murmur characteristics (timing, intensity and location) can occasionally discriminate between normal cats and cats with underlying cardiac disease, with the exception of loud (mostly systolic) and palpable murmurs inevitably associated with a significant heart disease. However, since the majority of heart murmurs in cats appear to be systolic and mild-moderate in loudness, echocardiography, ideally performed by an experienced cardiologist, should always be considered following identification of a heart murmur on routine physical examination in cats, especially when they do not present any clinical signs referable to a heart disease.

## Figures and Tables

**Figure 1 vetsci-09-00564-f001:**
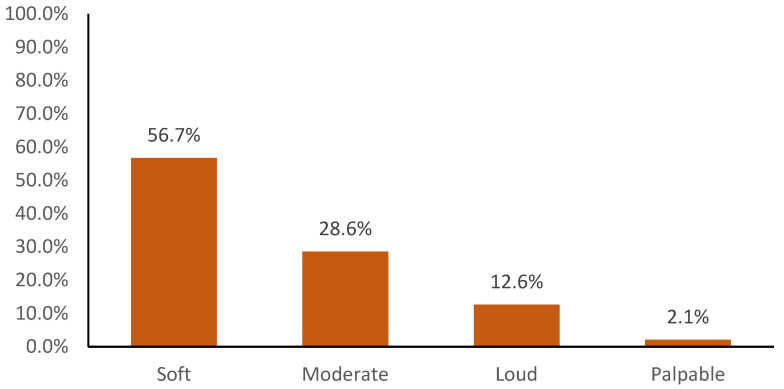
Histogram showing the prevalence of soft, moderate, loud and palpable heart murmurs in a population of 856 cats referred for diagnostic investigation of their murmur.

**Figure 2 vetsci-09-00564-f002:**
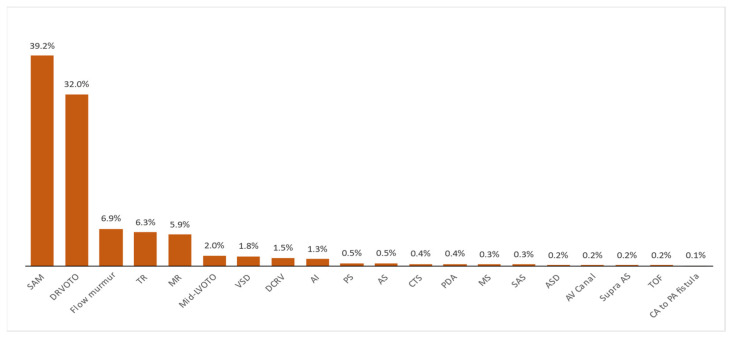
Histogram showing the frequency of various origins of blood flow turbulence responsible for the presence of an audible heart murmur in a population of 856 cats. SAM: systolic anterior motion of the mitral valve; DRVOTO: dynamic right ventricular outflow tract obstruction; TR: tricuspid regurgitation; MR: mitral regurgitation; Mid-LVOTO: mid-left ventricular outflow tract obstruction; VSD: ventricular septal defect; DCRV: double-chambered right ventricle; AI: aortic insufficiency; PS: pulmonic stenosis; AS: aortic stenosis; CTS: cor triatriatum sinister; PDA: patent ductus arteriosus; MS: mitral valve stenosis; SAS: sub-aortic stenosis; ASD: atrial septal defect; AV canal: atrioventricular canal defect; Supra AS: supravalvular aortic stenosis; TOF: tetralogy of Fallot; CA to PA fistula: coronary to pulmonary artery fistula.

**Table 1 vetsci-09-00564-t001:** List of breeds included in the study and their prevalence in the selected population of 856 cats with heart murmur.

Breed	Number	Percentage (%)
DSH	518	60.51
DLH	58	6.78
BSH	52	6.07
Bengal	49	5.72
Maine Coon	30	3.50
Persian	30	3.50
Ragdoll	16	1.87
Birman	16	1.87
Sphynx	16	1.87
Siamese	10	1.17
Siberian	9	1.05
British Blue	8	0.93
Scottish Fold	7	0.82
Burmese	6	0.70
Exotic	6	0.70
Norwegian Forest	4	0.47
Tonkinese	2	0.23
Savannah	2	0.23
Russian Blue	2	0.23
Devon Rex	2	0.23
Selkirk Rex	2	0.23
Chinchilla	2	0.23
Ocicat	2	0.23
Oriental shorthair	2	0.23
Himalayan	1	0.12
Egyptian Mau	1	0.12
Cyprus Shorthair	1	0.12
Korat	1	0.12
Turkish Van	1	0.12

DSH: domestic shorthair; DLH: domestic longhair; BSH: British shorthair.

**Table 2 vetsci-09-00564-t002:** Cause of the 1093 blood flow turbulences observed on echocardiographic study in 856 cats examined for investigation of heart murmur.

Cause	N	(%)	Notes
SAM	429	39.25	23 inducible
DRVOTO	350	32.02	71 inducible
Flow murmur	75	6.86	1 anemic
Tricuspid Regurgitation	69	6.31	
Mitral Regurgitation	64	5.86	
Mid-LVOTO	22	2.01	
Ventricular Septal Defect	20	1.83	
Double Chambered Right Ventricle	16	1.46	
Aortic Insufficiency	14	1.28	
Pulmonic Stenosis	6	0.55	
Aortic Stenosis	5	0.46	
Cor Triatriatum Sinister	4	0.37	
Patent Ductus Arteriosus	4	0.37	
Mitral Valve Stenosis	3	0.27	
Sub Aortic Stenosis	3	0.27	
Atrial Septal Defect	2	0.18	
Atrioventricular Canal Defect	2	0.18	
Supravalvular Aortic Stenosis	2	0.18	
Tetralogy of Fallot	2	0.18	
Coronary artery to Pulmonary artery fistula	1	0.09	

SAM: systolic anterior motion of the mitral valve; DRVOTO: dynamic right ventricular outflow tract obstruction; LVOTO: left ventricular outflow tract obstruction.

**Table 3 vetsci-09-00564-t003:** Result of multi-variable analysis showing the association between the probability of identifying a structural cardiac disease on echocardiographic examination and cat and heart murmur characteristics that were significant in the univariable analysis.

Variable	Coefficient	Odds Ratio	95% CI	*p* Value
Age	−0.01	0.99	0.96 to 1.02	0.5130
Soft murmur	−0.94	0.39	0.24 to 0.65	0.0003
Moderate murmur	−0.84	0.43	0.26 to 0.71	0.0010
Dynamic murmur	−0.35	0.70	0.49 to 1.02	0.0660
Intermittent murmur	−1.35	0.26	0.17 to 0.39	<0.0001
PMI = “RPA”	−0.69	0.50	0.32 to 0.78	0.0020

PMI: point of maximal intensity; RPA: right parasternal area.

## Data Availability

The data presented in this study are available on request from the corresponding author. The data are not publicly available due to patient data protection.

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
