# Peer review of "Prevalence and Clinical Significance of Heart Murmurs Detected on Cardiac Auscultation in 856 Cats"

_vetsci, 2022, doi:10.3390/vetsci9100564_

Round 1
Reviewer 1 Report
Good job, very useful research for practitioners
Author Response
The authors would like to thank Reviewer 1 for taking the time to review our manuscript and for the complimentary comments. Much appreciated.
Reviewer 2 Report
In the retrospective study the authors reviewed the medical records from 1521 feline heart mummer cases seen from June 2009 until June 2022 at two facilities. Of the 856 compete records the authors identified the identified source(s) of blood turbulence that may have contributed to that murmur.
The study aim/hypothesis are hard to identify. As such, the current manuscript version suffers from a lack of direction that challenges the review process.
It would seem that the authors are looking at sources of blood turbulence in cats with identified murmurs on examination. A second less well articulated aim (missing throughout the manuscript) is an attempt to correlate mummer characteristics with sources of turbulence. If this second aim is to be included, it needs to be introduced in the introduction (in the context of the literature) and addressed throughout the manuscript (and figures).
Additional information is provided, but the relevance of that information to the aim(s) of the manuscript are not addressed, considered or discussed (even as a limitation). Examples include animal demographic information (age, breed, sex) and the use of sedation during the examination. Each of these are a clear source of variability, which easily can be compared to determine if that variable contributed to the results. That analysis should be included. Alternatively, the potential significance of these variables (and body condition, and co-morbidities) should be discussed in the manuscript and study limitations section. In other words if animal age is important enough to mention (for example) than the effect of that variable needs to be explored, tested or at least discussed.
It is unclear why an accepted Levine murmur grading scale was reclassified to one more generic, less accepted and less useful. The authors are encouraged to revisit the heart mummer data (given that remains an aim of this paper) and look at correlation/associations between murmur timing, Levine grade, PMI and persistence.
The authors mention that the same stethoscope was used for the 13 year study duration, but not the same clinician doing the auscultation. This should more clearly be explained in the methods.
Despite this being a retrospective study, the authors provided considerable detail on the specifics of the procedure that were performed. That is confusing as it reads as if those procedures are being standardly performed as a part of this study. In fact, it can only be assumed that those exact procedures/tools were adhered to. The methods should be revised to focus on the methods that were performed as a part of this study only. Who and how they performed that review should be clearly explained.
The authors are cautioned to avoid over using “cause” throughout the study. Identifying a source of turbulence (in some cases one of multiple sources) is suggestive as a contributor to the patient’s mummer but each source of mummer may not be the cause. The authors are encouraged to consider to what degree the number of turbulent blood flow issues influenced the murmur characteristics investigated.
Identifying that significant cardiac abnormalities result in louder murmurs is not a novel finding. The authors should revise to indicate the study reinforces that observation rather than suggesting that finding as novel.
Figure 2 should be omitted, the information is repeated from table 2 (redundant) and is more difficult to read/interpret in the figure.
Author Response
The authors would like to thank Reviewer 2 for taking the time to review our manuscript and for the constructive comments. We have tried to address all the Reviewer’s comments and suggestions, as reported below:
Comments and Suggestions for Authors
In the retrospective study the authors reviewed the medical records from 1521 feline heart mummer cases seen from June 2009 until June 2022 at two facilities. Of the 856 compete records the authors identified the identified source(s) of blood turbulence that may have contributed to that murmur.
Q1 The study aim/hypothesis are hard to identify. As such, the current manuscript version suffers from a lack of direction that challenges the review process. It would seem that the authors are looking at sources of blood turbulence in cats with identified murmurs on examination. A second less well articulated aim (missing throughout the manuscript) is an attempt to correlate mummer characteristics with sources of turbulence. If this second aim is to be included, it needs to be introduced in the introduction (in the context of the literature) and addressed throughout the manuscript (and figures). Additional information is provided, but the relevance of that information to the aim(s) of the manuscript are not addressed, considered or discussed (even as a limitation). Examples include animal demographic information (age, breed, sex) and the use of sedation during the examination. Each of these are a clear source of variability, which easily can be compared to determine if that variable contributed to the results. That analysis should be included. Alternatively, the potential significance of these variables (and body condition, and co-morbidities) should be discussed in the manuscript and study limitations section. In other words if animal age is important enough to mention (for example) than the effect of that variable needs to be explored, tested or at least discussed.
A1 We would kindly disagree with Reviewer 2 and we personally believe that the aims of our study were clearly explained in the introduction. Furthermore, it is important to remember that most retrospective observational studies, even in human medicine, do not test a hypothesis. Indeed, unlike prospective studies, observations should offer a basis to generate a hypothesis at a later stage. Nevertheless, we would like to acknowledge Reviewer 2’s comment and we have expanded some sections, with a further clarification of the aims and including a paragraph explaining hypotheses (introduction) and how these were tested (Materials and Methods). As a consequence of these changes, a new statistical analysis (both univariable and multivariable analysis) has been performed. Finally, we would like to mention that the potential effect of various sedation protocols was already reported as a “limitation” in the original submission. Indeed, it would be almost impossible to reliably test this hypothesis with a statistical analysis due to the excessive number of variables, including type of drug/s used, doses, administration route, time of administration, environmental factors, etc, and the relatively low number of cats that were sedated in this study.
Q2 It is unclear why an accepted Levine murmur grading scale was reclassified to one more generic, less accepted and less useful. The authors are encouraged to revisit the heart mummer data (given that remains an aim of this paper) and look at correlation/associations between murmur timing, Levine grade, PMI and persistence.
A2. Once again, we would respectfully disagree with Reviewer 2. At least four recent peer-reviewed articles have addressed the limitations of the Levine system (inherited from human medicine without a proper validation) and proved the clinical utility of the new classification of heart murmur into the following categories (soft, moderate, loud and palpable):
- Ljungvall I, Rishniw M, Porciello F, Ferasin L, Ohad DG. Murmur intensity in small-breed dogs with myxomatous mitral valve disease reflects disease severity. J Small Anim Pract. 2014 Nov;55(11):545-50.
- Rishniw M. Murmur grading in humans and animals: past and present. J Vet Cardiol. 2018 Aug;20(4):223-233.
- Rishniw M, Caivano D, Dickson D, Swift S, Rouben C, Dennis S, Sammarco C, Lustgarten J, Ljungvall I. Breed does not affect the association between murmur intensity and disease severity in dogs with pulmonic or subaortic stenosis. J Small Anim Pract. 2019 Aug;60(8):493-498.
- Caivano D, Dickson D, Martin M, Rishniw M. Murmur intensity in adult dogs with pulmonic and subaortic stenosis reflects disease severity. J Small Anim Pract. 2018 Mar;59(3):161-166.
Q3 The authors mention that the same stethoscope was used for the 13 year study duration, but not the same clinician doing the auscultation. This should more clearly be explained in the methods.
A3. Thank you for this comment and please accept our apologies for the confusion. We meant the same “stethoscope model”. This has now been added in the text.
Q4 Despite this being a retrospective study, the authors provided considerable detail on the specifics of the procedure that were performed. That is confusing as it reads as if those procedures are being standardly performed as a part of this study. In fact, it can only be assumed that those exact procedures/tools were adhered to. The methods should be revised to focus on the methods that were performed as a part of this study only. Who and how they performed that review should be clearly explained.
A4. In our opinion, this comment is redundant. We have simply listed standard echocardiographic techniques and credentials used for echocardiographic classification of various cardiac abnormalities and we would expect all cardiologists worldwide to agree with such criteria. Furthermore, the accuracy of our observations has been double-checked retrospectively by a boarded cardiologist, as clearly explained in the text (“echocardiographic examinations were retrospectively reviewed by a board-certified cardiologist to double check the accuracy of measurements and diagnoses”).
Q5. The authors are cautioned to avoid over using “cause” throughout the study. Identifying a source of turbulence (in some cases one of multiple sources) is suggestive as a contributor to the patient’s mummer but each source of mummer may not be the cause. The authors are encouraged to consider to what degree the number of turbulent blood flow issues influenced the murmur characteristics investigated.
A5. We believe that even this comment is redundant. All heart murmur investigations performed by qualified cardiologists worldwide are aimed at identifying the cause of the murmur based on the presence of a significant blood flow turbulence on colour and spectral Doppler study. We agree that, in case of multiple turbulent flows, one turbulence may contribute to the genesis of the heart murmur more than others. However, in these scenarios, it would be simply impossible to prove whether one blood flow turbulence is the only cause of an audible murmur or if there is a combined contribution to the genesis of a murmur by multiple flow turbulences.
Q6. Identifying that significant cardiac abnormalities result in louder murmurs is not a novel finding. The authors should revise to indicate the study reinforces that observation rather than suggesting that finding as novel.
A6. We would be very grateful if Reviewer 2 could indicate some publications where such an association has been previously reported in cats, ideally in a large study like this.
Q7. Figure 2 should be omitted, the information is repeated from table 2 (redundant) and is more difficult to read/interpret in the figure.
A7. We agree that table 2 and figure 2 report the same results and may appear redundant. However, some readers find more comfortable to read results from figures rather than tables, and this is the reason why we left both on our manuscript. However, we are happy to remove one or the other should the Editor suggest to do so.
Reviewer 3 Report
The comments for Authors can be found in the attached PDF file.

Author Response
The authors would like to thank Reviewer 3 for taking the time to review our manuscript and for the useful and encouraging comments. We have tried to address all the Reviewer’s comments and suggestions, as reported below:
The manuscript describes the prevalence and the clinical significance of heart murmurs detected in a very large population of cats. The study’s idea as well as the topic of research are brilliant. Similarly, a great work has been done by the Authors to combine clinical and echocardiographic expertise. The findings described herein are interesting and clinically useful. Given the above, it was a great opportunity for me to Review this manuscript, and I congratulate with Author for their great job. Below some comments, questions and suggestions aimed at expanding further the results of the study and provide additional information to readers.
Q1. Simple Summary & Abstract
-Both in the Simple Summary and Abstract, the abbreviations “SAM”, “DRVOTO” have been introduced; however, these are used only twice in each section. Therefore, I am not sure the abbreviations could be maintained.
A1. Thank you for this comment. We are happy to remove these abbreviations if the Editor agrees.
2) M&M
Q 2.1 Data acquisition: among information that have been examined by Authors currently are: examination date, cat’s signalment, heart murmur characteristics, echocardiographic identification of blood flow turbulence (single or multiple) associated with the detected heart murmur, echocardiographic measurements, presence of structural abnormalities, ECG findings, and whether or not the cat underwent sedation prior to the echocardiographic examination.
Although the list of variables analyzed is wide, there are additional ones that should be taken into account given their potential effects on cardiovascular physiology and echocardiographic variables. For examples, I would re-examine the study population to see how many cats were hyperthyroid and/or anaemic cats, how many received corticosteroids, and how many cats received cardiovascular drugs (including antiarrhythmics). In the case of anaemic cats, I would also suggest to specify how many cats had a mild, moderate and severe anaemia, as a heart murmur due to anaemia is expected in severely anaemic cats. A last important clinical information concerning the hydration status of cats as this may influence the echocardiographic findings and also the heart murmurs (e.g. a severe volume depletion may exacerbate SAM and/or DRVOTO and or Mid-LVOTO). Therefore, I suggest to classify cats as normo-hydrated or affected by mild, moderate or severe dehydration (or a similar system of classification) using proper references. Lastly, I suggest to expand the statistical analysis to the association between these conditions and heart murmurs. I one or more of the abovementioned information is not obtainable retrospectively, this should be clearly stated among the limitations of the study.
A 2.1 We would like to thank Reviewer 3 for these comments. We agree that details of additional variables, such as blood pressure, thyroid status, haematocrit, administration of corticosteroids, and cardioactive drugs (including antiarrhythmics) would be extremely informative. Unfortunately, we were not able to retrospectively retrieve all this information and this important limitation was already disclosed in the manuscript.
Q 2.2 Heart murmur characteristic: I suggest to specify who made the cardiac auscultation (only a board certified cardiologist? Or a less expert veterinarian?). Such an information is important, as it has been demonstrated that clinicians with different levels of expertise on cardiac auscultation can obtain different results.
A 2.2 Thank you for this comment. We agree that a clarification was much needed and the appropriate changes have been made in the manuscript to clarify this point.
Q 2.4 Diagnostic criteria: for each echocardiographic abnormality I suggest to put a pertinent reference, so that readers could expand their interest on that particular malformation thanks the reference provided by Authors. This has been done, for example, for SAM, DRVOTO; MV dysplasia, Mid-LVOTO and Coronary artery to pulmonary artery fistula. However, references are lacking for PS, SAS/AS, DCRV, AI, VSD, ASD, ToF, PDA and. Atrioventricular canal defect. Moreover, concerning the reference used for Coronary artery to pulmonary artery fistula, I suggest to replace the one used by Authors with one or more from veterinary literature, since some references on this topic exist.
A 2.4 We would like to thank Reviewer 3 for this suggestion. Bibliographic citations have been added accordingly.
Q 2.6. Ethical approval: Authors stated that “Ethical approval was not sought due to the non-invasive and retrospective nature of this study.” Nevertheless, in many countries/Institutions, an internal approval approved by the Management Committee of the hospital is often required. Moreover, it would preferable if owners had given informed written consent for all the investigations. Please, specify is these have been obtained by Authors before study beginning/ publication.
A 2.6 This is correct and was already disclosed in the manuscript (“Written owner consent was obtained in all cases before each echocardiographic examination and, when deemed necessary, before any cat’s sedation”).
3) Results
Q 3.1 As said before, this section should be expanded saying how many cats were hyperthyroid, anaemic, the degree of anaemia, how many received corticosteroids, and how many were dehydrated and the severity of such dehydration. Moreover, I suggest to create a new table with cardiovascular drugs, including names and dosages.
A 3.1 Please see answer to comment 2.1
Q 3.2 Table 1 and Table 2: I suggest to control the numbers concerning percentage (%) reported in Table 1 and Table 2, as I am not sure that the sum is 100%.
A 3.2 We have double checked all data in table 1 and 2 and all numbers were correct. The sum was 99.8% (rather than 100%) because numbers were rounded up or down to maintain a single decimal for an easier reading. We have now added the second decimal to clarify this potential confusion.
4) General comments
Q 4.1 Abbreviations: I encourage Authors to recheck rules of the Journal and their use of abbreviation in the text as there are abbreviations that have been introduced but the not used so many times.
A 4.1 Thank you for this observation. We have tried to reduce the number of abbreviations wherever possible.
Q 4.2 Although the Authors made a great work trying to correlated echocardiographic findings with murmurs detection, I think that there is still a lacking information. The point is that, for example, if I have a cat with a mild DRVOTO and then I just say that there is a mild murmur without characterizing further its overall clinical condition and its type of murmur, then I could simply say that in that a case that murmur was due to DRVOTO. But this is not always necessarily the correct answer. Indeed, the cat may have for example, a severe anaemia in addition to mild DRVOTO, which may be the true cause of that a murmur. Then, in such a case, the murmur would be not due a cardiac trigger but to a systemic one. That’s why I strongly suggested authors to expand the number of information concerning the systemic condition of cats.
A 4.2 Thank you for this observation. As we explained below, we could not retrospectively retrieve all this clinical information and this important limitation was already disclosed in the manuscript. However, we would kindly disagree that some “hidden causes”, such as an anaemic status, could be deemed responsible for the genesis of a murmur. Ultimately, a heart murmur is always a blood flow turbulence by definition. The onset of a blood flow turbulence is regulated by the Reynold’s number. When the Reynold’s number exceeds a critical value, the blood flow shifts from laminar to turbulent flow (Fowler, N.O. (1991). Systolic Murmurs and Innocent Murmurs. In: Diagnosis of Heart Disease. Springer, New York, NY). In anaemic patients, the Reynold’s number (the predictor of turbulence) is increased due to decreased haematocrit (decreased viscosity). A second cause of turbulence in anaemic patients is a high cardiac output, which causes an increased blood flow velocity, which ultimately increases the Reynold’s number (Costanzo, L. S. (2018). Physiology [Sixth edition]. Philadelphia, PA: Elsevier). Therefore, if anaemia was the sole cause of a murmur, the cardiologist would expect to visualise a blood flow turbulence in the outflow tracts on colour and spectral Doppler echocardiography. Unfortunately, diagnosis of a flow murmur is rather challenging and is mainly based on the exclusion of any identifiable structural or functional cardiac abnormality (as explained in the text). We have reported one case of flow murmur in a confirmed anaemic patient but this does not provide a confirmation of the correct cause of the murmur. However, it would be difficult, from a practical point of view, to justify a blood sample to measure the patient’s haematocrit in every diagnosis of “flow murmur”.
Q 4.3 Moreover, to make more reliable the results of the study, it should be made an addition analysis. It should be see if, in each cat with a murmur and a possible echocardiographic justification for that a murmur, the murmur characteristics really supported the hypothesized association. I try to explain better this with an example. If you just say that you have a cat with a murmur and SAM, then it is straight to say that the murmur was due to SAM. But if you try also to characterize the murmur, then such an association does not necessarily exist. Indeed, if you have a cat with an apical right-sided systolic murmur or a sternal diastolic murmur and SAM on echo, that that a murmur cannot be related to SAM. Therefore, I suggest to expand the results section to show the type of murmurs among cats with each type of cardiac defect. For this purpose, you can also create a new Table, where you can say: “among cats with SAM, XX had a left apical systolic murmur, XX a sternal systolic murmur, XX had no audible murmurs…..Among cats with DRVOTO XX, has a sternal systolic murmur, XX had no audible murmurs….etc”.
A 4.3 In principle, we would agree with this suggestion. However, the point of maximal intensity depends on several factors, including, but not limited to, body position, chest conformation, direction of the turbulence, etc, which are not necessarily easy to evaluate. We personally believe that this additional analysis may generate confusion rather than providing valuable information for the clinical activity.
Reviewer 4 Report
This is an excellent study examining the clinical significance of the murmur on a large number of cats. The article can be published in the veterinary science journal after perform some minor revisions, especially on the statistical analysis, listed below:
Materials and methods, lines 189-204 (statistical analysis): the intensity of the murmur (grade 1 to 6) is a qualitative variable and as such must be examined. Qualitative variables cannot be expressed as mean and standard deviation. The authors do not specify that they reported the results of the qualitative variables by percentage as evident in the results section.
Results
In the numerical results eg 5.6 (5.2-6.1) years (line 216) the authors do not specify what the values in brackets are: are they the confidence intervals or the interquartile range values? Please specify.
Lines 218-219: authors should indicate the % of neutered animals out of the total of males and females.
I suggest indicating the respective numerical value for each result in % as done in line 307.
Lines 308-311: this sentence is unclear.
Lines 320-321: Authors should report significance (p value) and association coefficient.
From the table 3 it appears that all murmur types are significantly associated with the probability of detecting structural abnormalities on echocardiography because all the p values are less than 0,05.
Lines 424-428: in the M&M section (lines 85-86) the authors stated that all examinations were performed by 2 veterinarians and that all cardiac auscultations were performed with a single stethoscope (lines 82-83). Please check.
Author Response
The authors would like to thank Reviewer 4 for taking the time to review our manuscript and for the useful and encouraging comments. We have tried to address all the Reviewer’s comments and suggestions, as reported below.
This is an excellent study examining the clinical significance of the murmur on a large number of cats. The article can be published in the veterinary science journal after perform some minor revisions, especially on the statistical analysis, listed below:
Q 1 Materials and methods, lines 189-204 (statistical analysis): the intensity of the murmur (grade 1 to 6) is a qualitative variable and as such must be examined. Qualitative variables cannot be expressed as mean and standard deviation. The authors do not specify that they reported the results of the qualitative variables by percentage as evident in the results section.
A 1.Thank you for this comment. Please note that the intensity of the murmur in this manuscript was analysed using the system proposed by Rishniw (Journal of Veterinary Cardiology (2018) 20, 223-233). We did not apply the Levine system for the reasons explained in the text.
Results
Q 2 In the numerical results eg 5.6 (5.2-6.1) years (line 216) the authors do not specify what the values in brackets are: are they the confidence intervals or the interquartile range values? Please specify.
A 2. The values in brackets indicate 95% CI for the median. This was explained in Methods and Materials (2.5 Statistical analysis).
Q 3 Lines 218-219: authors should indicate the % of neutered animals out of the total of males and females.
A 3. This was already reported in the text (Results, 3.1.1. Animals)
Q 4. I suggest indicating the respective numerical value for each result in % as done in line 307.
A 4. We are not sure about this comment. All results in table 1 and 2 report numerical values and the relative percentage.
Q 5. Lines 308-311: this sentence is unclear.
A 5. We are sorry to learn that Reviewer 4 found the sentence unclear. The sentence states: “However, when these types of murmurs were analyzed individually as a sole cause of murmur, 56.1% of cats with SAM and 85.0% of cats with DRVOTO did not display any structural cardiac abnormalities on echocardiographic examination”. This means that when SAM or DRVOTO are identified alone and not in combination with other flow turbulences (i.e. only one blood flow turbulence identified on echocardiography), the percentage of cats without identifiable structural abnormalities increases. We hope this clarifies the confusion.
Q 6. Lines 320-321: Authors should report significance (p value) and association coefficient.
A 6. Thank you for this comment. Since all palpable, continuous, diastolic and to-and-fro murmurs were always associated with a significant congenital cardiac disease (100%), any statistical analysis would be superfluous and overall redundant (i.e. there are not any other possible outcomes).
Q 7. From the table 3 it appears that all murmur types are significantly associated with the probability of detecting structural abnormalities on echocardiography because all the p values are less than 0,05.
- 7 Only “soft” and “moderate” heart murmurs, those exhibiting a dynamic or intermittent behaviour and those with a PMI over the right parasternal area present a significant negative association with the presence of a cardiac abnormality on echocardiography (multivariable analysis). However, table 3 has now been changed to accommodate the request of another reviewer who asked to include additional variables.
Q 8. Lines 424-428: in the M&M section (lines 85-86) the authors stated that all examinations were performed by 2 veterinarians and that all cardiac auscultations were performed with a single stethoscope (lines 82-83). Please check.
A 8. Thank you for this comment and please accept our apologies for the confusion. We meant the same “stethoscope model”. This has now been added in the text.
Round 2
Reviewer 2 Report
The original version of this manuscript was strong with only a few minor revisions recommended to the authors (by the other reviewers and me) before acceptance could be recommended. These comments are meant to ensure that clear quality research is published. I was very taken aback by the authors replies and lack of receptivity to reviewer comments. At first, I thought this was just in response to my review but discovered that unfortunate trend in response to the other reviewers as well. If a topic is of question or concern, it is best addressed with additional clarity in the manuscript. Often a single statement can be made in the paper to explain or justify a choice.
For example, the authors use of the term “cause” throughout the manuscript is in reference to the clinical etiology. To prove causation scientifically one must determine X came before Y, that the relationship between the two (X and Y) did not occur by chance and that nothing else accounts for the relationship between X and Y. The authors did not establish causation for many of the claims that they made so I asked that an alternative phrase be used to clarify. The authors could have used etiology, correlation or made a statement to clarify their definition in the manuscript. Instead, the authors rejected the suggestion outright.
The authors also differed to the editors stating that they would only make a revision if the editor required it. I am surprised that authors are so unwilling to make these minor adjustments that they would call the editor in to make these decisions. That puts additional time and effort on the editor’s part who already sought out content experts to make their suggestions. If these editor-mediated-decisions are not brought explicitly to the editor’s attention, these corrections can be missed.
I still feel that this manuscript has a great deal of potential. I cannot however recommend acceptance without addressing my and the other reviewers’ comments more completely. I would also suggest that the authors refrain from critiquing and providing commentary on the reviewer comments as that can be counter productive. I hope that the authors can find ways to address the aforementioned concerns by all of the reviewers in a may that produces the best quality research publication.
Author Response
We have been asked by the Editor to reply only to Reviewer 3 and Reviewer 4's comments
Reviewer 4 Report
I thank the authors for making some of the requested changes. However, some errors remain in the description of the statistical analysis and in the way in which the data are summarized in the results section:
Statistical analysis:
Lines 215-218: the murmur intensity is a qualitative variable (using either the Levine or the Rishniw classification), as such it cannot be summarized as mean ± standard deviation or median ± 95% confidence interval as written by the authors. Moreover, the median is a measure of central tendency of quantitative variables which, in descriptive statistic, is usually reported together with the range or the interquartile range and not with the 95% confidence intervals (in descriptive statistic it is not necessary to report 95% CI).
The authors should specify how they summarized the qualitative variables. Qualitative variables are usually reported using numbers and/or percentages.
In my opinion the sentence "Data were analyzed for normal distribution using the Shapiro–Wilk test for age, bodyweight, and murmur intensity and subsequently reported as mean ± standard deviation for normally distributed or median ± 95% confidence interval (CI) for non-normally distributed data.” should be replaced by “Quantitative variables were analyzed for normal distribution using the Shapiro – Wilk test and subsequently reported as mean ± standard deviation if normally distributed or median and range if non-normally distributed. Qualitative variables were summarized using absolute and relative frequencies. "
Results section:
I suggest summarizing all the quantitative variables, not normally distributed, as median (range)
I suggest reporting all the qualitative variables as numbers and percentages (n=…;..%)
